# Psychometric Calibration of a Tool Based on 360 Degree Videos for the Assessment of Executive Functions

**DOI:** 10.3390/jcm12041645

**Published:** 2023-02-18

**Authors:** Francesca Borgnis, Francesca Borghesi, Federica Rossetto, Elisa Pedroli, Luigi Lavorgna, Giuseppe Riva, Francesca Baglio, Pietro Cipresso

**Affiliations:** 1IRCCS Fondazione Don Carlo Gnocchi ONLUS, 20148 Milan, Italy; 2Department of Psychology, University of Turin, 10124 Turin, Italy; 3Applied Technology for Neuro-Psychology Lab, IRCCS Istituto Auxologico Italiano, 20149 Milan, Italy; 4Faculty of Psychology, eCampus University, 22060 Novedrate, Italy; 5Division of Neurology, University of Campania Luigi Vanvitelli, 80131 Naples, Italy; 6Humane Technology Lab, Catholic University of the Sacred Heart, 20123 Milan, Italy

**Keywords:** executive functions, 360° environments, virtual reality, convergent validity, psychometric assessment, 360° videos

## Abstract

Introduction: Over the last decades, interactive technologies appeared a promising solution in the ecological evaluation of executive functioning. We have developed the EXecutive-functions Innovative Tool 360° (EXIT 360°), a new instrument that exploits 360° technologies to provide an ecologically valid assessment of executive functioning. Aim: This work wanted to evaluate the convergent validity of the EXIT 360°, comparing it with traditional neuropsychological tests (NPS) for executive functioning. Methods: Seventy-seven healthy subjects underwent an evaluation that involved: (1) a paper-and-pencil neuropsychological assessment, (2) an EXIT 360° session, involving seven subtasks delivered by VR headset, and (3) a usability assessment. To evaluate convergent validity, statistical correlation analyses were performed between NPS and EXIT 360° scores. Results: The data showed that participants had completed the whole task in about 8 min, with 88.3% obtaining a high total score (≥12). Regarding convergent validity, the data revealed a significant correlation between the EXIT 360° total score and all NPS. Furthermore, data showed a correlation between the EXIT 360° total reaction time and timed neuropsychological tests. Finally, the usability assessment showed a good score. Conclusion: This work appears as a first validation step towards considering the EXIT 360° as a standardized instrument that uses 360° technologies to conduct an ecologically valid assessment of executive functioning. Further studies will be necessary to evaluate the effectiveness of the EXIT 360° in discriminating between healthy control subjects and patients with executive dysfunctions.

## 1. Introduction

Neuropsychological assessment is historically considered an integral part of the neurological examination and consists of the normatively informed application of performance-based assessments of various cognitive skills [1]. Among these cognitive abilities, the evaluation of executive functioning represents a challenge for neuropsychologists, due to the complexity of the construct [2] and the methodological difficulties [3,4,5].

The executive functions involve a wide range of neurocognitive processes and behavioral skills (e.g., reasoning, decision making, problem solving, planning, attention, control inhibitor, cognitive flexibility, and working memory) that appear to be crucial in many real-life situations [6]. Their dysfunction, typical in psychiatric and neurological pathologies, constitutes a significant global health challenge, due to their high impact on personal independence, social abilities (e.g., work, school, relationships), and cognitive and psychological development [7,8,9]. Specifically, executive function deficits affect daily tasks such as meal preparation, money management, housekeeping, and shopping [10,11], with an inevitable impact on the person’s quality of life and feelings of personal well-being [12]. Moreover, subjects with executive function impairments show problems in starting and stopping activities, increased distractibility, difficulties in learning, generating or planning strategies, and difficulties with online monitoring and inhibiting irrelevant stimuli [13,14]. Thus, identifying early strategies for evaluating and rehabilitating these disorders appears to be a priority [15].

The executive functions are traditionally evaluated with standard paper-and-pencil neuropsychological tests such as the Modified Wisconsin Card Sorting Test [16], Stroop Test [17], Frontal Assessment Battery [18,19], or the Trail Making Test [20], which allow standardized procedures and scores that make them valid and reliable. However, several studies have demonstrated that these tests were not able to predict the complexity of executive functioning in real-life settings [4,6,21,22,23,24]. An ecological assessment of executive functions appears critical for achieving excellent executive dysfunction management [23], given the significant impact of executive functions on daily life and personal independence [11,25]. Therefore, innovative neuropsychological tests have been developed aimed at evaluating executive functioning within real-life scenarios [3], such as the Multiple Errands Test (MET) [21], in which patients are assessed while they are carrying out shopping tasks in a real supermarket, or UCSD Performance-based Skills Assessment (UPSA-B), in which patients must perform everyday tasks in two areas of functioning: communication and finances [26]. Data showed that these ecological evaluations provided a more accurate estimate of the patient’s deficits than were obtained within the laboratory [27]. However, they showed several limitations, including extended times, high economic costs, the difficulty of organization, poor controllability of experimental conditions and poor applicability for patients with motor deficits [28].

Therefore, the ecological limitations of the traditional neuropsychological battery and difficulties in administering tests in real-life scenarios have led researchers and clinicians to search for innovative solutions for achieving an ecologically valid evaluation of executive functions. In this framework, the use of interactive technologies (e.g., virtual reality, serious games, and 360° video) appeared as a promising solution, because they simulate real environments, situations, and objects, thereby allowing an ecologically valid assessment of executive functions [29,30,31,32] with a rigorous control on principal variables [33,34,35]. Several studies have shown VR-based tools to be appropriate instruments for assessing and rehabilitating executive functions, because they allowed clinicians to evaluate subjects while performing several everyday tasks in ecologically valid, secure, and controlled environments that reproduce everyday contexts [36,37,38]. Moreover, these VR-based instruments guarantee good control of the perceptual environment, a precise stimuli presentation, greater applicability, and user-friendly interfaces, and enable the acquisition of data and analysis of performance in real-time [14,31,32,33,39,40,41]. Indeed, several studies have shown the efficacy of VR-based assessment tools of executive functions in neurological and psychiatric populations, showing impairments invisible to traditional measurements [27,32,39,42,43,44,45,46,47,48].

In recent years, one of the most promising trends in the VR technology field is 360° technology [49], which has appeared as an interesting instrument in different healthcare sectors, including neuropsychological assessment [49,50,51], rehabilitation [52], and educational training [53]. Specifically, in neuropsychological assessment, the advances in 360° technologies allowed participants to be evaluated in virtual environments (photographs or immersive videos), which they experience from a first-person perspective without particular clinical negative effects (e.g., nausea, vertigo), enhancing the global user experience of evaluation [32]. In this direction, Serino and colleagues have developed a 360° version of the Picture Interpretation Test (PIT) for the detection of executive deficits (only the active visual-searching component), which has been successfully tested on patients with Parkinson’s disease and multiple sclerosis [51,54]. Following these promising results, Borgnis and colleagues developed the EXIT 360° (Executive-functions Innovative Tool 360°) for gathering information about many components of executive functioning (e.g., planning, decision making, problem solving, attention, and working memory) [55]. Indeed, the EXIT 360° was born to provide a complete evaluation of executive functionality, involving participants in a ‘game for health’ delivered via smartphones, in which they must perform everyday subtasks in 360° environments that reproduce different real-life contexts. Two previous studies showed promising and interesting results regarding usability, user experience, and engagement with the EXIT 360° in healthy control subjects [56] and patients with Parkinson’s disease [57]. Participants had a positive global impression of the tool, evaluating it as usable, easy to learn to use, original, friendly, and enjoyable. Interestingly, the EXIT 360° also appeared to be an engaging tool, with high spatial presence, ecological validity, and irrelevant adverse effects.

Since our purpose in developing the EXIT 360° is to produce an innovative tool for evaluating executive functions, this work aimed to assess the convergent validity of the EXIT 360°. The concept of ‘convergent validity’ means how closely the new scale or tool relates to other variables and measures of the same construct [58]. In other words, it assumes that tests based on the same or similar constructs should be highly correlated. In this context, one of the most used methods is to correlate the scores between the new assessment tool and others claimed to measure the same construct [59]. For this purpose, we have compared the EXIT 360° with standardized traditional neuropsychological tests for executive functioning.

## 2. Materials and Methods

### 2.1. Participants

Seventy-seven healthy control subjects were consecutively recruited at IRCCS Fondazione Don Carlo Gnocchi in Milan, according to the following inclusion criteria: (a) aged between 18 and 90 years; (b) education ≥ 5 years; (c) absence of cognitive impairment as determined by the Montreal Cognitive Assessment test [60] (MoCA score ≥17.54, cut-off of normality), corrected for age and years of education according to Italian normative data [61]; (d) absence of executive impairments as evaluated by a traditional neuropsychological battery for executive functioning; (e) ability to provide written, signed informed consent. Exclusion criteria: (a) overt hearing or visual impairment or visual hallucinations or vertigo; and (b) systemic, psychiatric, or neurological conditions.

The study was approved by the Fondazione Don Carlo Gnocchi ONLUS Ethics Committee in Milan. All participants obtained a complete explanation of the study’s purpose and risk before filling in the consent form, based on the revised Declaration of Helsinki (2013).

### 2.2. Procedure of Study

All participants underwent a one-session evaluation that involved three main phases: a pre-task evaluation (1), followed by an EXIT 360° session (2), and a brief post-task evaluation (3) [62].

#### 2.2.1. Pre-Task Evaluation

Before the study’s initiation, the subjects signed the written informed consent and completed a questionnaire designed to collect the participants’ demographic data (e.g., age, gender, education level). Then, participants underwent an evaluation using traditional paper-and-pencil neuropsychological tests to exclude the presence of frank deficits in global cognitive and executive functioning. In detail, the neuropsychological evaluation allowed assessment of participants’ compliance with the inclusion criteria, and the convergent validity between the traditional neuropsychological tests for executive functions and the EXIT 360°.

The global cognitive profile was investigated using the MoCA test, a sensitive screening tool to exclude the presence of cognitive impairment.

Moreover, the neuropsychological battery for executive functioning included: Trail Making Test (in two specific sub-tests: TMT-A and TMT-B) [20], Phonemic Verbal Fluency Task (F.A.S.) [63], Stroop Test [17], Digit Span Backward [64], Frontal Assessment Battery (FAB) [18,19], Attentive Matrices [65] and Progressive Matrices of Raven [66,67,68]. Table 1 gives a detailed description of the different executive functions evaluated by each of these neuropsychological tests.

#### 2.2.2. EXIT 360° Session

After the neuropsychological assessment, all participants underwent an evaluation with the EXIT 360°. The neuropsychologist started the administration by inviting participants to sit on a swivel chair and wear a mobile-powered headset. Before wearing the headset, the psychologist provided a specific general instruction: “*You will now wear a headset. Inside this viewer, you will see some 360° rooms of a house. To visualize the whole environment, I ask you to turn yourself around; you are sitting on a swivel chair for this reason. Within these environments, you will be asked to perform some tasks*”.

The EXIT 360° consists of 360° immersive domestic photos as virtual environments in which participants have to perform a preliminary familiarization phase and seven subtasks of increasing complexity.

In the first phase (one minute), participants had to familiarize themselves with the technology (they had to explore a 360° neutral environment freely) and report any side effects (such as nausea and vertigo) by answering ad-hoc questions (“*Did you feel nauseous and/or dizzy during the exploration?*”). If adverse effects occurred (of any intensity), the examiner had to stop the test immediately. Otherwise, subjects were immersed in a 360° environment representing a living room.

The real session (and time registration) started when the participants heard the following instruction: “*You are about to enter a house. Your goal is to get out of this house in the shortest time possible. To exit, you will have to complete a path and a series of tasks that you will find along your way. Are you ready to start*?”. During the EXIT 360° session, participants are immersed in several virtual environments that they must explore simply by moving their heads and rotating themselves, while remaining seated on the swivel chair.

Participants had to leave the house by completing the domestic path in the shortest possible time, while overcoming all seven subtasks: (1) Let’s Start; (2) Unlock the Door; (3) Choose the Person; (4) Turn On the Light; (5) Where Are the Objects?; (6) Solve the Rebus; and (7) Create the Sequence (for a detailed description, see [55,62]) (Figure 1).

Briefly, the seven subtasks reproduce everyday scenarios that ask the subject to solve specific assignments according to the instructions. Let’s Start requires participants to observe a map and choose the path that allows them to reach the ‘finish’ in the shortest possible time. In the second subtask, the subjects have to open a door choosing between three specific options: key, telephone, and drill. The Choose the Person task requires the participant to explore a living room and select the correct person according to a particular instruction. In task 4, the subjects are immersed in a dark room because ‘the power went out,’ and they have to choose an object (i.e., flashlight) that allows them to continue the journey. In the following task, participants must identify the piece of furniture (among several pieces) on which four specific objects are placed. In task 6, subjects must complete a rebus consisting of many tiles, each containing a number and a geometric shape of different colors. Next to these tiles, a blank tile is inserted containing two question marks for the subject to fill in. Finally, they must memorize a sequence of numbers in the last task, reporting them in reverse.

Overall, the subtasks are designed to evaluate different components of executive functioning (e.g., planning, decision making, divided attention), and their level of complexity changes according to the cognitive load and the presence of confounding variables. Table 2 shows an overview of executive functions that the seven EXIT 360° subtasks could evaluate.

To respond to subtasks’ requests, the subjects had to choose between three or more options, which allowed them to solve the task in the best possible way. Interestingly, in the mobile-powered headset, participants saw a small white dot/square, a ‘pointer’ that follows their gaze. When participants wanted to answer within the environment, they had to move their head, positioning the white dot over the answer for a few seconds. The response was then selected automatically.

The psychologist recorded all the subjects’ responses and reaction times: participants received only one point for the wrong answer (vs two for a correct one).

The digital solution was implemented to allow the subject to proceed automatically across the subtasks when they answered correctly. Where there was an error, the system provided visual feedback to the patient: “*You have obtained a score of 1; inform the investigator*”. Moreover, response times were calculated from the end of each subtask instruction until the participant provided the answer.

Overall, the EXIT 360° allowed data to be collected about a participant’s total score (range 7–14), and subtask and total reaction times (i.e., time in seconds registered from examiner’s instruction until the participant provided the last correct answer).

#### 2.2.3. Post-Task Evaluation

After the EXIT 360° session, all subjects rated the usability of the EXIT 360° through the System Usability Scale [69,70,71,72], a ten-item questionnaire on a five-point scale, from ‘completely disagree’ to ‘strongly agree’, with the total score (range 0–100) indicating the overall usability of the system.

### 2.3. Statistical Analysis

Descriptive statistics included frequencies, percentages, median and interquartile range (IQR) for categorical variables, and mean and standard deviation (SD) for continuous measures. Skewness, Kurtosis, and histogram plots were visually explored to check the variables’ normal distributions, and perform parametric or non-parametric analyses when adequate. Pearson’s correlation (or Spearman’s correlation) was applied to evaluate the possible relationship between the scores of neuropsychological tests and the EXIT 360° (total score and subtask scores). Moreover, Pearson’s correlation was conducted to compare the total EXIT 360° score with the usability score. Furthermore, we evaluated the association (with univariate and multiple linear regression) between EXIT 360° variables and demographic characteristics to verify the possible influence of socio-demographic features on the results of the innovative tool. All statistical analyses were performed using Jamovi 1.6.7 software. A statistical threshold of *p* < 0.05 on two-tailed tests was considered statistically significant.

## 3. Results

### 3.1. Participants

Table 3 reports the demographic and clinical characteristics of the whole sample. The subjects (*n* = 77) are predominantly female (M:F = 29:48), with a mean age of 53.2 years (SD = 20.40, range = 24–89), and mean years of education of nearly 13 (IQR = 13–18, range 5–18). All participants included in the study showed an absence of cognitive impairment (MoCA correct score = 25.9 ± 2.62).

### 3.2. Traditional Neuropsychological Assessment

Table 4 reports the mean scores (raw and corrected scores) of neuropsychological tests with the respective cut-off of normality (equivalent score ≥ 2). All participants in the study showed scores within the normal range on all traditional neuropsychological tests for executive functions.

### 3.3. EXIT 360°

All participants in the study completed the whole task, obtaining only one point for wrong answers or two points for correct ones. Figure 2 reports participants’ scores (%) on all seven subtasks.

Overall, the descriptive analysis showed that healthy controls obtained a total score of 12.6 (±1.02; range = 10–14), with 88.3% of subjects receiving a score of ≥12. Regarding the total reaction time, participants took about 8 min (mean = 480 s ±130 sec; range = 192–963 sec) to complete the whole task.

The univariate linear regression shows a significant impact from age (β = −0.451, *p* < 0.001; R2 = 0.203) and education (*p* < 0.001; R2 = 0.300) on the EXIT 360° total score, but not from gender (β = −0.0980; *p* = 0.680; R2 = 0.002). Specifically, regarding education, a significant difference emerges between a low level of education (5 years) and medium to high ones, respectively 13 (β = 1.635, *p* < 0.001), 16 (β = 1.962, *p* < 0.001) and 18 (β = 1.923, *p* < 0.001). Moreover, Pearson’s correlation showed a significant negative correlation between age and total score (r = –0.451; *p* < 0.001). Regarding the EXIT 360° total reaction time, univariate linear regressions showed no significant impact from all of the demographic characteristics on the time variable (*p* > 0.05). The multiple linear regression (R2 = 0.342) confirmed the effect of education on the EXIT 360° total score (*p* < 0.05), but not the impact of age, which showed only a tendency to significance (β = −0.239, *p* = 0.051). Finally, the variable ‘sex’ did not impact the EXIT 360° total score (β = −0.127, *p* = 0.528).

### 3.4. Correlation between Neuropsychological Tests and EXIT 360°

Table 5 shows the correlations (Pearson’s correlation) between the traditional paper-and-pencil neuropsychological tests and the two scores of the EXIT 360°.

Specifically, Pearson’s correlation showed a significant correlation between the EXIT 360° total score and all neuropsychological tests. Moreover, data showed a correlation between the EXIT 360° total reaction time and several tests, particularly the timed ones (e.g., Trail Making Test, Stroop Test, and Attentive Matrices).

Furthermore, data showed no relationship between the EXIT 360° total score and EXIT 360° reaction time (*p* = 0.587), and only a correlation between the score and time of task 4 (r = 2.31; *p* < 0.05).

Finally, Table 6 shows the significant correlation between traditional neuropsychological tests and the seven subtask scores (Spearman’s correlation) and reaction time (Pearson’s correlation).

### 3.5. Usability

The mean value of the usability, calculated with the SUS, was 75.4 ± 13.2, indicating an acceptable level of usability, according to the scale’s score (cut-off = 68) and adjective ratings (Figure 3).

Specifically, according to the cut-off score (cut-off = 68), more than 70% of participants showed scores above the cut-off. In addition, according to the adjective rating, 35.5% of subjects evaluated the EXIT 360° as ‘Good’, 32.9% as ‘Excellent’, and 27.6% as ‘Best imaginable’ [71].

Pearson’s or Spearman’s correlation showed no significant linear correlation between the SUS total score and the demographic characteristics, particularly for age (r = −0.045, *p* = 0.699) and education (r = −0.096; *p* = 0.405). Moreover, data showed the absence of a significant correlation between the SUS total score and the total score of the EXIT 360° (r = 0.126; *p* = 0.276).

## 4. Discussion

Over the last few years, there has been a growing interest in using VR-based solutions for making an ecologically valid assessment of executive functioning in several clinical populations [37,38,51,54]. Indeed, many studies have shown the efficacy of VR-based tools in evaluating executive functions in neurological and psychiatric populations, showing impairments invisible to traditional measurements [42,51].

Borgnis and colleagues used the advance of 360° technologies to develop the EXIT 360°, an innovative assessment instrument that aims to detect several executive deficits quickly, involving participants in a ‘game for health’, in which they must perform everyday subtasks in 360° environments that reproduce different real-life contexts [55,62]. After evaluating the usability of the EXIT 360° in a healthy control sample [56] and subjects with a neurological condition [57], the authors have assessed the convergent validity of this innovative tool for assessing executive functionality, comparing it with traditional standardized neuropsychological tests for executive functioning. Indeed, it is well known that a strong positive correlation between a new tool and other instruments designed on the same construct is evidence of the high convergent validity of the new test [59].

Findings on usability confirmed previous research, demonstrating a good-to-excellent usability score, with over 32% of participants evaluating the EXIT 360° as excellent and 27.6% as the best imaginable [56]. Interestingly, data showed no correlation between the total usability score and the total score of the EXIT 360°. Therefore, the score obtained by the participants in our innovative 360°-based tool is not influenced by the usability level, but only by participants’ performance (as also highlighted by the correlation between neuropsychological tests and total score).

Moreover, our data to test convergent validity showed a significant correlation between the EXIT 360° total score and all neuropsychological tests for executive functioning. Furthermore, an interesting and promising association was found between the EXIT 360° total reaction time and timed neuropsychological tests, such as the Trail Making Test, Stroop Test, and Attentive Matrices. As previously mentioned, a high correlation between the indexes of the new test (EXIT 360°) and the scores of other standardized instruments that evaluate the same construct (i.e., executive functioning) supported the high level of convergent validity of the new tool [59]. Therefore, we can conclude that the EXIT 360° showed a good convergent validity. In other words, the EXIT 360° can be considered an innovative solution to evaluate several components of executive functioning: selective and divided attention, cognitive flexibility, set shifting, working memory, reasoning, inhibition, and planning [55].

Further analysis showed that an evaluation with the EXIT 360° did not require a long administration time; indeed, participants took, on average, about 8 min to complete the entire task. This finding suggests that the EXIT 360° can be considered a quick and useful screening instrument to evaluate executive functioning. As regards the accuracy score, that is, the EXIT 360° total score, most of the participants (over 88%) achieved high scores (≥12). Further studies will be conducted to determine if a score of 12 could be a good cut-off value, able to differentiate between healthy and pathological groups.

In addition, no difference appeared in both EXIT 360° scores due to gender. Moreover, no impact of age and education appeared on the time variable. On the contrary, a difference occurred between the low education level (5) and medium-to-high education level groups on the EXIT 360° total score. A relationship also appeared between age and total score, with the older participants obtaining low scores. However, considering the joint impact of the demographic characteristic on the EXIT 360° total score results confirmed only the effect of the education variable (with only a tendency to significance for the age variable). As a result, just as with most neuropsychological tests, it will be necessary to provide a standardization of total scores for age and education.

Other analyses were conducted on the seven tasks to evaluate the performance at each task (correct answers), and any correlation between them and neuropsychological tests. These further analyses aimed to determine the: (1) potential differences in the complexity of subtasks and (2) executive functions evaluated by each. Data showed that Task 7 was the most complex (only 64.2% of participants gave a correct answer), followed by Task 6 (66.2%). Except for Task 1, the correlation analysis showed a growing cognitive load across the tasks. Moreover, the EXIT 360°, with its seven subtasks, appeared as a valuable and promising ecologically valid instrument to assess: [a] selected and divided attention (subtasks 3–5–6–7), [b] cognitive flexibility (subtasks 1–4–7–6–7), [c] inhibition control and interference sensitivity (subtasks 1–4–6–7), [d] working memory (subtasks 1–5–6–7), [e] planning (subtasks 4–6–7), [f] visual search (3–7), [g] set switching (subtask 1–5–6), and [h] reasoning (subtask 5–6–7). These findings supported the rationale behind the concept and design of the EXIT 360° activities, built to increase in terms of cognitive load (number of cognitive components evaluated) during the 360° experience. However, introducing confounding variables (distractors) could also increase the difficulty. Indeed, Tasks 5 and 6 assess the same load of executive functions, but Task 6 appeared more complex in terms of percentages of correct answers (90 vs. 66.2), due to the addition of confounding variables.

As regards Task 2, no correlation appeared with neuropsychological assessment; however, this result is not surprising since Task 2 was developed to evaluate the decision making that was not measured by the selected tests. As a result, the introduction to the neuropsychological evaluation of a test to measure decision-making ability could demonstrate the capacity of Task 2 to assess this executive function. Moreover, an additional possible explanation could be the ‘ceiling effect’, as all control subjects performed the task correctly.

Since executive function is a complex and heterogeneous construct with a high impact on everyday life and personal independence [11,25], an ecological evaluation of more components of executive functioning appears crucial to achieve optimal executive dysfunction management [23].

Despite the promising results, the present work has some limitations. Firstly, this study did not evaluate the inter-rater reliability assessment of the EXIT 360° for reaction time accuracy. Secondly, the neuropsychological tests chosen for convergent validity do not allow for assessing all components of executive functioning (for example, decision making). Another limitation is represented by the potential session-order effect. We think that the evaluation sessions of executive functions are based on different methodological paradigms (digital function led vs. conventional paper-and-pencil tests). However, another study may use a protocol in which the order of administration of the sessions is randomized to test the potential effect of session order.

## 5. Conclusions

This work appears as a first validation step towards considering the EXIT 360° as a valid and standardized instrument that exploits 360° technologies for conducting an ecologically valid assessment of executive functioning. Further studies will be necessary to: (1) provide standardization of the EXIT 360° total score for age and education, (2) assess the EXIT 360° inter-rater and test-retest reliability, to deepen its potential as a new screening tool; (3) evaluate the effectiveness of the EXIT 360° in discriminating between healthy control subjects and patients with executive dysfunctions; and (4) implement an automated scoring system for response times.

## Figures and Tables

**Figure 1 jcm-12-01645-f001:**
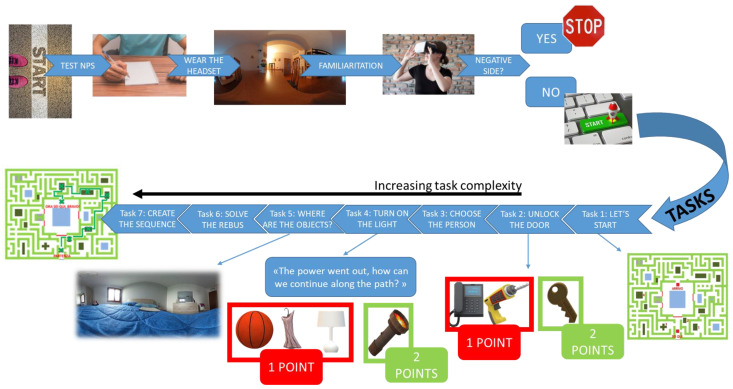
Evaluation process.

**Figure 2 jcm-12-01645-f002:**
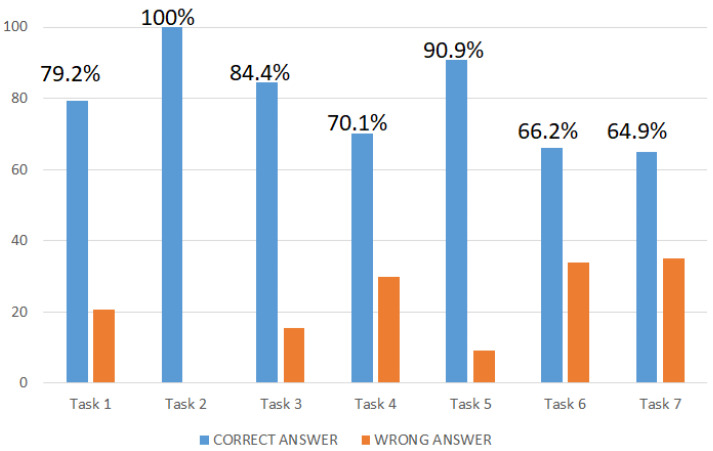
Scores (%) on seven subtasks.

**Figure 3 jcm-12-01645-f003:**
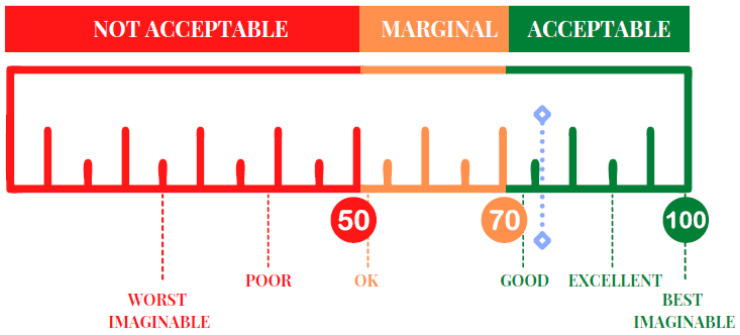
A graphic representation of the SUS score.

**Table 1 jcm-12-01645-t001:** Pre-task evaluation: Neuropsychological tests.

Name	Executive Function
Trail Making Test	Visual search Task switching Cognitive flexibility
Verbal Fluency Task	Access to vocabulary on phonemic key
Stroop Test	Inhibition
Digit Span Backward	Working memory
Frontal Assessment Battery	Abstraction Cognitive flexibility Motor programming/planning Interference sensitivity Inhibition control
Attentive Matrices	Visual search Selective attention
Progressive Matrices of Raven	Sustained and selective attention Reasoning

**Table 2 jcm-12-01645-t002:** EXIT 360° subtasks and related executive functions.

	Name	Executive Function
Task 1	Let’s Start	Planning–Inhibition Control–Visual Search
Task 2	Unlock the Door	Decision Making
Task 3	Choose the Person	Divided Attention–Inhibition Control–Visual Search
Task 4	Turn On the Light	Problem Solving–Planning–Inhibition Control
Task 5	Where Are the Objects?	Visual Search–Selective and Divided Attention–Reasoning
Task 6	Solve the Rebus	Planning–Reasoning–Set shifting–Selective and Divided Attention
Task 7	Create the Sequence	Working Memory–Selective Attention–Inhibition Control

**Table 3 jcm-12-01645-t003:** Demographic and MoCA scores of the whole sample.

		Subjects [*n* = 77]
**Age** (years)	*Mean (SD)*	53.2 (20.40)
**Sex** (M:F)		29:48
**Education** (years)	*Median (IQR)*	13 (13–18)
**MoCA_raw score**	*Mean (SD)*	26.9 (2.37)
**MoCA_correct score**	*Mean (SD)*	25.9 (2.62)

M = Male; F = Female; SD = Standard deviation; IQR= Interquartile range; *n* = Number; MoCA = Montreal Cognitive Assessment.

**Table 4 jcm-12-01645-t004:** Scores of neuropsychological assessment.

Neuropsychological Tests	Raw Score *Mean (SD)*	Corrected Score *Mean (SD)*	Cut-Off of Normality
Trail Making Test–Part A *	37.2 (22.9)	35.1 (19.3)	≤68
Trail Making Test–Part B *	94.1 (58.9)	90.6 (48.4)	≤177
Trail Making Test–Part B-A *	56.9 (42.3)	57 (34.2)	≤111
Verbal Fluency Task	41.6 (11.1)	37.9 (9.46)	≥23
Stroop Test–Errors	0.68 (1.09)	0.62 (1.13)	≤2.82
Stroop Test–Time *	22.6 (15.5)	23.8 (11.5)	≤31.65
Frontal Assessment Battery	17.6 (1)	17.7 (0.85)	≥14.40
Digit Span Backward	4.77 (0.99)	4.56 (0.97)	≥3.29
Attentive Matrices	54.3 (5.53)	48.6 (6.43)	≥37
Progressive Matrices of Raven	32.3 (3.63)	32.3 (3.2)	≥23.5

SD = Standard Deviation; * = Total time in seconds.

**Table 5 jcm-12-01645-t005:** Correlation between EXIT 360° scores and neuropsychological assessment. In bold, statistically significant scores. * *p* < 0.05; ** *p* < 0.001.

	EXIT 360° Total Score	EXIT 360° Total Reaction Time
Montreal Cognitive Assessment	**0.48** **	**−0.31** *
Progressive Matrices of Raven	**0.44** **	**-**
Attentive Matrices	**0.26** *	**−0.23** *
Frontal Assessment Battery	**0.41** **	**-**
Verbal Fluency Task	**0.54 ****	**-**
Digit Span Backward	**0.32** *	**-**
Trail Making Test–Part A	**-**	**0.14**
Trail Making Test–Part B	**-**	**0.27** *
Trail Making Test–Part B-A	**-**	**0.29** *
Stroop Test–Errors	**−0.32** *	**0.25** *
Stroop Test–Time	**−0.45** **	**0.28** *

**Table 6 jcm-12-01645-t006:** Correlation between subtask scores and neuropsychological assessment. Correlation significances are represented by colors: blue = not statistically significant scores; yellow = scores tending to statistical significance; orange = *p* < 0.05; red = *p* < 0.001.

	Task 1		Task 2		Task 3		Task 4		Task 5		Task 6		Task 7	
	Score	Time	Score	Time	Score	Time	Score	Time	Score	Time	Score	Time	Score	Time
**PMR**	n.s.	n.s.	n.s.	n.s.	n.s.	n.s.	n.s.	n.s.	0.241	n.s.	0.484	n.s.	0.296	n.s.
**AM**	n.s.	n.s.	n.s.	n.s.	n.s.	−0.218	n.s.	n.s.	n.s.	n.s.	n.s.	n.s.	0.284	−0.226
**FAB**	n.s.	n.s.	n.s.	n.s.	n.s.	n.s.	0.254	n.s.	n.s.	n.s.	0.266	n.s.	0.283	n.s.
**V.F.T.**	n.s.	n.s.	n.s.	n.s.	n.s.	n.s.	n.s.	n.s.	n.s.	n.s.	0.489	n.s.	0.438	n.s.
**DS**	n.s.	−0.269	n.s.	n.s.	n.s.	n.s.	n.s.	n.s.	0.251	n.s.	0.341	−0.253	0.303	n.s.
**TMT–A**	n.s.	n.s.	n.s.	n.s.	n.s.	n.s.	n.s.	n.s.	−0.301	n.s.	−0.462	n.s.	−0.299	0.244
**TMT–B**	n.s.	0.333	n.s.	n.s.	n.s.	n.s.	n.s.	n.s.	−0.31	n.s.	−0.36	n.s.	n.s.	n.s.
**TMT B-A**	n.s.	0.366	n.s.	n.s.	n.s.	n.s.	n.s.	n.s.	−0.259	n.s.	n.s.	n.s.	n.s.	n.s.
**ST_E**	n.s.	n.s.	n.s.	n.s.	n.s.	n.s.	n.s.	n.s.	−0.29	0.28	n.s.	n.s.	n.s.	n.s.
**ST_T**	n.s.	0.339	n.s.	n.s.	n.s.	n.s.	−0.282	n.s.	−0.297	n.s.	−0.344	n.s.	−0.329	0.286

PMR = Progressive Matrices of Raven; AM = Attentive Matrices; FAB = Frontal Assessment Battery; VFT = Verbal Fluency Task; DS = Digit Span Backward; TMT-A = Trail Making Test–Part A; TMT-B = Trail Making Test–Part B; TMT-B-A = Trail Making Test–Part B-A; ST_E = Stroop Test–Errors; ST_T= Stroop Test–Time.

## Data Availability

Data can be obtained upon reasonable request to the corresponding author.

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
