# Peer review of "Psychometric Calibration of a Tool Based on 360 Degree Videos for the Assessment of Executive Functions"

_jcm, 2023, doi:10.3390/jcm12041645_

Round 1

Reviewer 1 Report

Peer Review for J of Clin Med of Exit 360 Psychometric Calibration

To the Authors:

The study reported in this manuscript is part of a series of papers on a VR approach to measuring executive functioning.  The authors should be commended on the project overall and their efforts to proceed to scientifically validate their innovative methods. What follows is a review by sections but I want to comment that there are many very minor but annoying (to an English reader) English errors and that a native English speaker might be useful to make sure that the text does not contain these minor errors. I will point them out as I go through each section.

Abstract: Line 25: with the 88.3% that obtained … should be with 88.3% obtaining a high total score.

Line 26: data released … I think it should be data revealed.

I appreciate that the abstract does not overstate the findings.

Introduction: I liked the introduction which made a logical and compelling case for the need for new EF instruments.  Absent from the paragraph beginning with line 75 is any mention of the UPSA-B (Mausbach, B. T., Harvey, P. D., Pulver, A. E., Depp, C. A., Wolyniec, P. S., Thornquist, M. H., ... & Patterson, T. L. (2010). Relationship of the Brief UCSD Performancebased Skills Assessment (UPSAB) to multiple indicators of functioning in people with schizophrenia and bipolar disorder. Bipolar disorders12(1), 45-55.).  It’s been used in schizophrenia and HIV research.

Line 113 and Line 117 contain references that appear to be cut and paste.  They are not formatted correctly and don’t appear in the references.

English problems: Line 50 inevitably .. should be inevitable

Line88 In recent years, one of the most trends.  Word missing. “promising trends”?

Line 111: Since EXIT 360 wants to be… I don’t think you should be personifying the test. Better, Since our purpose in developing EXIT 360 is to produce an innovative..

Materials and Methods

Line 124 I am surprised that the Italian cut-off for MoCA is 17.54.  That is much lower than the North American version and the findings of the sample which was raw score of 26.9, which is consistent with North American norms. Please just check it.

Table 1 is confusing with out lines separating each task.  For example, I am sure digit span backwards is not about cognitive flexibility and that the descriptor probably belongs with FAB.

Line 160.  I know these instructions are a translation but “turn on yourself” is not the correct English translation.  I assume you mean “turn yourself around.”

Line 167, On the contrary, doesn’t make sense here.

Paragraph beginning 179.  I know the authors reference a previous paper with longer descriptions of the tasks, but Task 6 is only described as a rebus and that is not informative.  This really matters since Task 6 is the mostly strongly correlated with the NPS tests. One general question for me was whether the VR environment just gamifies a standard problem-solving task. Task 7 is described as a kind of digit span backward task and later shows a r = .30 with DS.

Line 201: Apparently the task is hand scored by a psychologist.  The word “instead” doesn’t make sense here. More importantly, there is no information about what the scorer is seeing that is being scored.  Presumably, they are watching on a screen what the participant is doing and recording the time and accuracy, but this should be described.  Although the scoring seems straightforward, there is no mention of any inter-rater reliability for either the performance score or the accuracy of the reaction times. This needs to be addressed.  If it is a weakness of the study, it needs to be in a limitations section. I am a little surprised that there is no mention of automated scoring given the sophistication of the technology. If that is part of the development plan, it would be useful to say something about this in the discussion.

Line 222: The statistical threshold does not mention if this is a one tailed or two-tailed test.

Line 230: the title of Table 2 states Demographic and Clinical Characteristics.  Since only the MoCA score is offered, it is really not a table that is characterizing clinical characteristics.  After all, it is a normal sample.  Better would be Demographics and MoCA scores

Line 247: There is a very wide range for RT and no SD is offered or any statement about whether this was normally distributed.  This matters since correlations appear to be Pearson’s which assume normality.  This needs to be addressed.

Discussion:

On the whole, I think the authors have fairly represented their findings.

Line 333: The authors appear to be making a claim that their instrument can “allow an assessment of multiple components of EF in a short time. That is not what they found at all and they need to rewrite or remove this claim.  They have demonstrated that the total score has convergent validity with a number of tasks.  It is a screening measure (which they say in the next sentence).  They certainly have some indication that certain tasks are more strongly related to specific EF’s but that is very different from claiming that they can derive from their findings a specific assessment of attention and distinguish it from working memory or response inhibition from cognitive flexibility.  Indeed, one might think that they will want to be reconsidering what tasks they wish to use given that at least in a normal sample, the first three tasks are just adding noise (that is uncorrelated with EF) to their total score and tasks 2,4,5 and 6 have almost no relationship with speed. 

Line 338: will be conducted to deepen.  I believe they mean to determine

Line 344: the older participants that obtained low scores. I believe it should read with the older participants obtaining lower scores.

Line 354 a growing cognitive load in the tasks.  I believe they mean across the tasks.

Line 360-362: This needs to be rewritten.  I am not sure what they are saying here.

There is no Limitations section and they must add that.  There are many limitations including the tasks they chose for convergent validity, the racial and ethnic limitations of the sample, the reliability of scoring (if not addressed), and there is a question about whether they should have created an EF composite score or EF factor with the convergent measures and correlate such a score with their measure.  This would eliminate concerns about experiment-wise error from multiple comparisons.

Conclusions: No concerns, although they might have expanded on their next steps a little mor

Reviewer 2 Report

This work seeks to validate a VR method (EXIT360) developed by the authors by comparison with traditional neuropsychological tests. Such validation is required for further use of EXIT360 in clinical/psychological studies. Such application development will definitely advance the field of neuropsychological assessment, making those tests more flexible to use, more accessible potentially without requiring supervision.

I have a few concerns which should be convincingly addressed by the authors before acceptance for publication:

1. In the design of the study, the participants follow the exact same order of tests for evaluation: 1) the pre-task evaluation consisting of conventional assessment using a list of NPS then 2) the EXIT360 session and finally 3) the post-task evaluation consisting of scoring the usability of EXIT360 using the SUS scoring system.

Is there a rationale for the order of the tests? The authors do not consider nor discuss a potential session-order effect which could be a confounding factor. Can the pre-task session have an effect on the results observed for EXIT360 session? A randomization of the test order would have avoided such potential bias.

The same issue concerns intra-session tasks. The authors state that EXIT360 tasks present an increasing difficulty in executive function. Is this statement based on the success rate of participants? Is it possible that there is an effect of fatigue instead of difficulty which could explain lower results for task 6 and 7?

Are the NPS provided during the pre-evaluation also of increasing difficulty?

These aspects should be better discussed.

2. The authors mentioned that during the familiarization step at the beginning of phase 2, they ask the participants to “report any side effects. If adverse effects occurred, the examiner had to stop the test immediately”. What was considered “adverse effect”? Intensity? This is not a scientific approach of measuring adverse effects induced by VR. There are multiple short and reliable tests available to measure cybersickness such as VRISE test (5 questions) or SSQ (16 questions). Authors should detail in section 2.2.2 the side effects and the correspponding intensity they considered for study withdrawal.

3. The authors measured Reaction Time. However, the use of this term is confusing and rather corresponds to the total duration of the EXIT360. Reaction Time corresponds to a measure of the time taken by a participant to respond/achieve a task. Unless the authors have measured the time spend to solve each of the 7 tasks individually after the questions were displayed, they should replace Reaction Time by Completion Time or Time to Complete EXIT360.

Moreover this reviewer has some difficulty to understand the rationale of the correlation of EXIT360 Total Reaction Time with the NPS (Table 4). Some of the NPS are timed some are not. Reaction time of the timed NPS is not presented. What justifies the calculation of a correlation between EXIT360 Total Reaction Time and non-timed NPS tests?

Furthermore, an interesting and promising association was found between EXIT 360°, Total Reaction Time and timed neuropsychological tests, like the Trail Making Test, Stroop Test, and Attentive Matrices” (l. 321-322) is not a satisfactory explanation. A rationale/justification for calculating correlation for Reaction Time must be clearly mentioned.

4. Table 5  presents the correlation between EXIT360 subtasks scores and NPS. Subtasks 2 and 3 do not seem to correlate to any of NPS. This is not addressed in the discussion. Are these tasks relevant and will they be conserved in EXIT360?

5. The NPS assessments are based on cut-off values (Table 3)  indicating whether the participants are in the normal range or not. Here all participants are within the normal range. EXIT360 provides measures (number of correct answers, time to completion). A significant proportion of participants gave wrong answers (between 0% and 35.1% depending on the subtask). How will cut-off values of EXIT360 results corresponding to normality be determined?

Minor concerns:

1. It is unclear to this reviewer whether participants following the EXIT360 session do have to actually walk to exit the virtual environment or is it purely a static task. This should be clearly mentioned in the description of the EXIT360.

2. The authors sort out the NPS according to the assessment of the corresponding executive function (Table 1). For better readability, this table should be completed with the 7 EXIT360 tasks which can be matched to the corresponding executive function (appearing l. 351 – 360). This would help future readers to identify the specificity of each task in EXIT360.

Round 2

Reviewer 2 Report

Thanks to the authors for addressing my concerns. I am satisfied with their explanations and subsequent modifications in the manuscript, except for my concern #1. My question might have been unclear, so I will rephrase. I questioned here a potential session-order effect. In other terms, would the results of EXIT360 and NPS be the same if the order would have been switched: NPS-EXIT360-EXIT360 usability = EXIT360-EXIT360 usability-NPS?

I understand NPS was used for 2 purposes: compliance with inclusion criteria + convergent validity. However, there can still be a effect of the NPS on the EXIT360 outcomes. The authors should consider this potential bias and discuss it.

Author Response

Thanks for this clarification. We added and discussed that the chosen NPS-EXIT 360°-EXIT 360° usability sequence could lead to a potential session-order effect and, thus, a potential bias.